

# A critical review of challenges and opportunities for effective design and operation of offshore structures supporting green hydrogen production, storage, and transport

Claudio A. Rodríguez, Baran Yeter, Shen Li, Feargal Brennan, Maurizio Collu

Department of Naval Architecture, Ocean & Marine Engineering, University of Strathclyde, 100 Montrose Street, Glasgow G4 0LZ, United Kingdom

*Correspondence to*: Claudio A. Rodriguez (claudio.rodriguez-castillo@strath.ac.uk)

**Abstract.** The climate emergency has prompted rapid and intensive research into sustainable, reliable, and affordable energy alternatives. Offshore wind has developed and exceeded all expectations over the last two decades and is now a central pillar

of the UK and other international strategies to decarbonise energy systems. As the dependence on variable renewable energy resources increases, so does the importance of the necessity to develop energy storage and non-electric energy vectors to ensure a resilient whole-energy system, also enabling difficult to decarbonise applications, e.g., heavy industrial, heat and certain areas of transport. Offshore wind and marine renewables have enormous potential that can never be completely utilised by the electricity system, and so green hydrogen has become a topic of increasing interest.

Although numerous offshore and marine technologies are possible, the most appropriate combinations of power generation, materials and supporting structures, electrolysers and support infrastructure and equipment depend on a wide range of factors, including the potential to maximise the use of local resources. This paper presents a critical review of contemporary offshore engineering tools and methodologies developed over many years for the upstream oil & gas, maritime and more recently offshore wind and renewable energy applications and examines how these along with recent developments in modelling and

digitalisation might provide a platform to optimise green hydrogen offshore infrastructure. The key-drivers and characteristics of future offshore green hydrogen systems are considered, and a Strengths, Weaknesses, Opportunities, and Threats (SWOT) analysis provided to aid the discussion of the challenges and opportunities for the offshore green hydrogen production sector.

**Keywords**: Ocean renewable energy, offshore wind, offshore structures, green hydrogen

| ABBREVIATIONS AND NOMENCLATURE | |
| --- | --- |
| O&G – Oil and Gas | TLP – Tension Leg Platforms |
| OTEC – Ocean Thermal Energy Conversion | SLS – Serviceability Limit State |
| OWT – Offshore Wind Turbine | FLS – Fatigue Limit State |
| LCoE – Levelised Cost of Energy | ULS – Ultimate Limit State |
| LCoH2 – Levelised Cost of Hydrogen | SHM – Structural Health Monitoring |





| | |
|---|---|
| PEM – Proton Exchange Membrane Electrolysis | LIDAR – Laser Imaging, Detection, and Ranging |
| AEL – Alkaline Electrolysis | AI – Artificial Intelligence |
| AC/DC – Alternating Current/Direct Current | SWOT – Strength, Weakness, Opportunity, and Threat |
| CH2 – Compressed Hydrogen | kW – Kilowatt |
| LH2 – Liquefied Hydrogen | MW – Megawatt |
| LOHC – Liquid Organic Hydrogen Carrier | \$/kgH2 – United States Dollar per kilogram of hydrogen |
| CAPEX – Capital Expenditure | €/kgH2 – Euro per kilogram of hydrogen |
| TRL – Technology Readiness Level | kW/m² – Kilowatt per square metre |
| DNV – Det Norske Veritas | kW/m³ – Kilowatt per cubic metre |
| ABS – American Bureau of Shipping | mm/year –millimetre per year |
| IEC – International Electrotechnical Commission | $D/t$ – Diameter-to-thickness ratio |
| API – American Petroleum Institute | |

## 1 Introduction

The climate emergency has prompted research, development, and commercialization of new renewable energy technologies on an unprecedented scale. Offshore wind in Northern Europe has to date become a tremendous success surpassing most commercial and technical expectations. This success has buoyed investor and political confidence and led to even greater
ambitions for sustainable electricity generation. Such a large proportion of electricity from a variable renewable energy resource, however, highlights the importance of energy storage to ensure a reliable and resilient whole energy system. Local storage of energy can address the necessity to transport electricity over longer distances and using alternative energy vectors such as hydrogen and ammonia can help manage the output energy variability, as the stored energy can be fed back into the grid when needed. In addition, it can offer the potential to make more sustainable those sectors difficult to decarbonise, such
as renewable heat, heavy industrial and transport demand applications.

Ocean renewable energies have a vast, yet barely developed potential, that includes not only the most known sources such as wind, wave, and tides, but also others such as solar, thermal, or chemical (e.g., salinity gradient) resources. Numerous technologies have been investigated and developed to varying degrees, but only a few have reached a maturity level to allow commercial development.

The following sections review pertinent studies that may potentially support offshore hydrogen production and associated functions such as storage and transport from an offshore support structure perspective. Rather than directly consider current offshore wind technology as the default solution for offshore hydrogen production as in (Apostolou and Enevoldsen, 2019; D'amore-Domenech and Leo, 2019; Dinh et al., 2021; Jang et al., 2022), our approach is to perform critical analyses to identify challenges and opportunities based on the knowledge and experience from closely related sectors such as offshore oil and gas



(O&G) and offshore wind applications. Fundamental scientific and engineering understanding for the conversion of ocean renewable energy to liquid and/or gaseous fuels is set out, so that optimal offshore structural solutions can be developed. Challenges and opportunities considered are those specifically related to "green" fuel production, including storage and transportation, and not to those already addressed for offshore wind electricity generation such as in (Asim et al., 2022; Bashetty and Ozcelik, 2021; Leimeister et al., 2018b; Otter et al., 2021).

**2 Overview of hydrogen systems for offshore deployment**

**2.1 Concepts proposed**

In recent years, several offshore green hydrogen demonstration projects have been reported, the most prominent being ERM Dolphin (Caine et al., 2021), the Vattenfall project (Vattenfall, 2022), and Lhyfe (Rem, 2022), but much earlier concepts were proposed in the 1970s using ocean thermal energy conversion (OTEC) devices (Dugger and Francis, 1977). The use of offshore

wind energy to produce alternative fuels followed years later, e.g. Dutton (2003); Kassem (2003); Steinberger-Wilckens (2002); Young et al. (1975). The majority of these studies considered the fuel production onshore from electricity generated offshore. Murahara and Seki (2008) pioneeringly at the time proposed offshore wind energy integrated with hydrogen production within a single offshore facility. A "Megafloat" offshore electrolysis plant that utilised offshore wind energy to produce sodium as solid fuel to then obtain hydrogen was proposed. The system concept was presented together with estimates

of power required to operate the plant and of the amounts of chemical by-products that could be commercialised. A similar concept for the floating structure was also proposed in (Tsujimoto et al., 2009), designed as a "sailing" wind farm. The offshore wind farm accommodated hydrogen production equipment along with a storage capacity of 9,900 tonnes of hydrogen in the form of organic hydride.

The proposed use of more conventional arrangements of offshore wind turbines (OWT) to produce "green" fuels has since

been reported by several authors. Most use the terms "stand-alone" or "integrated" to refer to offshore wind generation connected to a hydrogen production plant located onshore and powered by local, non-grid connected electricity. Zhao et al. (2011) was found to be the earliest study of a remote offshore wind energy system that stores energy in the form of hydrogen and chlorine. Additionally, a dynamic analysis model that considered the variability of wind energy production to address the performance of hybrid wind-hydrogen systems was proposed. The thermodynamic energy balance, the energy management

strategy, the performance of the wind energy conversion, the electrolyser array, the modified PEM (Proton Exchange Membrane Electrolysis) fuel array, and the hydrogen storage systems were also detailed.

With the success of onshore wind turbines, and more recently offshore wind, an increasing number of papers concerning offshore wind to hydrogen have been published. Some of the most relevant and prominent are detailed in references (Rogeau et al., 2023; Lee et al., 2023; Kumar et al., 2023; Bonacina et al., 2022; Ibrahim et al., 2022; Henry et al., 2022; Jang et al.,

2022; Mehta et al., 2022; Lucas et al., 2022; Dinh et al., 2021; Franco et al., 2021; Song et al., 2021; Woznicki et al., 2020; Caglayan et al., 2019; Hou et al., 2017; Ioannou and Brennan, 2019; Serna et al., 2015; Meier, 2014). These are mainly



concerned with: techno-economic aspects of wind-hydrogen systems, feasibility studies, sensitivity analyses of costs, different scenarios for hydrogen production, storage and transportation, comparisons between distributed, centralised, and onshore hydrogen production, comparisons of electrolyser technologies, use of alternative energy vectors for hydrogen exportation

such as ammonia and liquid organic hydrogen carriers, modelling and simulations of energy management or control strategies for the generated energy, the balance between produced hydrogen and consumed energy, and other related topics related to the Levelised Cost of Hydrogen (LCoH2).

In addition to offshore wind turbines, alternative concepts/technologies to harness wind energy to produce hydrogen have also been proposed and developed. In Schmitz and Madlener (2015), a fully automated "Power Ship" towed by a so-called "Sky

Wing" (a kite) was proposed in which energy could be stored in the form of compressed air or hydrogen. An economic feasibility analysis concluded that compared to stationary floating wind, this concept avoids the need for foundations or mooring systems. It is suggested that maintenance and repair can be done in sheltered harbour/coastal areas and has mobility potential to locate to more favourable sites depending on seasonal/other factors. Gilloteaux and Babarit (2017b) proposed a wind-driven catamaran dedicated to hydrogen production using Flettner rotors to propel a ship and hydro-generators to convert

the kinetic energy of the water flow in electricity and then, to hydrogen by electrolysis (Fig. 1). The numerical case studies demonstrated that an eighty metre long catamaran could produce net power of almost 500 kW at a forward speed of 8 m/s. Clodic et al. (2018) not dissimilarly investigated a wind energy ship concept dedicated to hydrogen production by analysing the advantages and limitations of four different technologies: rigid sails, kite wings, air foils and Flettner rotors and concluded that all the selected technologies are capable of converting wind energy into kinetic energy and then hydrogen. Based on those

findings, Babarit et al. (2018) reports the techno-economic feasibility of fleets of wind-driven sailing ships and platforms for far offshore hydrogen production. Different scenarios for hydrogen production, transportation and (compressed or liquified) storage were assessed. Due to the low volumetric energy density of hydrogen at standard temperature and pressure, the high costs of transportation and distribution were identified as the most challenging. In Babarit et al. (2019), other fuel options such as synthetic natural gas, methanol, Fisher-Tropsch fuel and ammonia were considered and concluded that in terms of

energy and economic performance, methanol may be the best energy vector for the far offshore wind energy ships systems, despite the costs associated with $CO_2$ production. Another alternative but not dissimilar concept for offshore hydrogen production and onboard storage from wind energy was proposed in Alexander (2019) (Fig. 2).



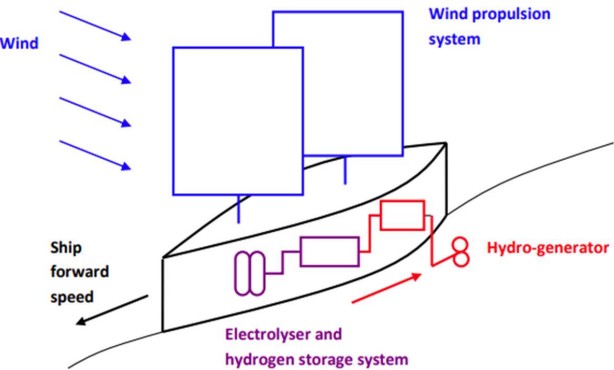

**Figure 1: Working principle of a wind-driven ship dedicated to hydrogen production. Reproduced from Gilloteaux and Babarit**
**(2017a) with permission from The American Society of Mechanical Engineers.**

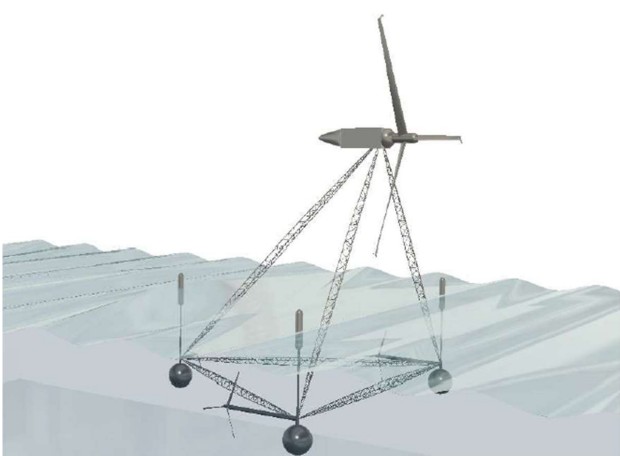

**Figure 2: Alternative concept of an offshore wind energy–hydrogen integrated production system. Reproduced from Alexander (2019) with permission from The American Society of Mechanical Engineers.**


From the literature review reported above, it can be seen that the majority of research to date in this area has unsurprisingly focused upon the techno-economics of offshore wind–hydrogen systems and in particular on whether a centralised or distributed system is likely to be optimal. Offshore wind turbines on fixed bottom support structures are currently the most mature technology for harnessing offshore wind energy and probably the best option for integration of hydrogen production

in shallow waters in the near future, however, larger wind energy deployments are taking place in deeper waters where floating



systems are the only available option. Given this development, it would seem prudent that further investigations and analyses need to be performed not only in terms of thermodynamics, energy generation and consumption associated with hydrogen production but also on the stability, sea- and station- keeping performance of potential floating support structures. The sizing and adaptation of conventional offshore marine energy support structures to allow containing hydrogen production equipment,
storage and offloading facilities has yet to be addressed. In contrast to offshore wind facilities for electricity generation where fixed monopiles and jacket structures are broadly accepted for shallow water turbine support and semisubmersibles, spars and barges seem to be the preferred options for deep waters, further research needs to be conducted to study which support structural configuration better match the requirements of offshore hydrogen production systems.

### 2.2. System configuration strategies

A number of studies have analysed and compared possible typologies or scenarios for the hydrogen system configuration, i.e., onshore, centralised offshore, distributed offshore, shared with O&G platforms, partially (surplus energy) or fully dedicated. For example in Jepma et al. (2018), the net present economic value for the following scenarios was analysed:
   a) Offshore wind energy farm plus shared hybrid O&G platform that includes electrolysers and export pipelines and cables.
b) Offshore wind farm plus shared O&G platform or direct connection to the grid.
   c) Hydrogen production is integrated into each wind turbine (distributed hydrogen production) plus the O&G platform (for hydrogen compression and offloading).

It was concluded that locating electrolysers together with wind turbines (a distributed hydrogen scenario) is likely to be economically superior to other alternatives due to significant cost savings in AC/DC conversions and cables for electricity
transportation. Jang et al. (2022) also compared the net present value among three configurations for the connection of wind power plants with hydrogen production facilities, namely, centralised onshore, distributed offshore and centralised offshore. Based on Rogeau et al. (2023); Ibrahim et al. (2022); Jang et al. (2022), the following configurations are envisaged for generic ocean renewable hydrogen system facilities:

**a) Centralised onshore** in which the ocean renewable energy farm produces electricity which is sent ashore powering the
hydrogen plant. Assuming a remote location, it is required to gather the AC electricity output from each ocean energy device to a substation to be converted to DC so that it can be sent ashore via submarine cables. Once onshore, a new electricity conversion from DC to AC is likely to be needed to power the electrolyser plant (Fig. 3a).

**b) Centralised offshore** is similar to Centralised Onshore, but with the hydrogen production on an offshore large-scale electrolysis floating platform near the ocean renewable energy farm. The hydrogen production platform is likely to contain
desalination units, electrolysers, cooling units, compression units (depending on the type of electrolyser), a hydrogen buffer tank, and a back-up battery system. The electricity from each of the ocean energy devices would be delivered via electric cables to an offshore substation where the total electricity is combined to power the centralised hydrogen production platform. The produced hydrogen can then be exported (Fig. 3b).




**c) Distributed (decentralised) offshore** means that the equipment necessary for hydrogen production is accommodated by

each individual ocean energy device. A typical layout for the electrolysis facility is likely to include a desalination unit, an

electrolyser, a cooling unit, a compression unit (if PEM electrolyser is not adopted), a hydrogen storage tank (as a buffer) and

a battery storage system as back-up power for the facility. The individual (hydrogen) production from each device is delivered

through risers and can be collected in a subsea manifold to be exported (Fig. 3c).

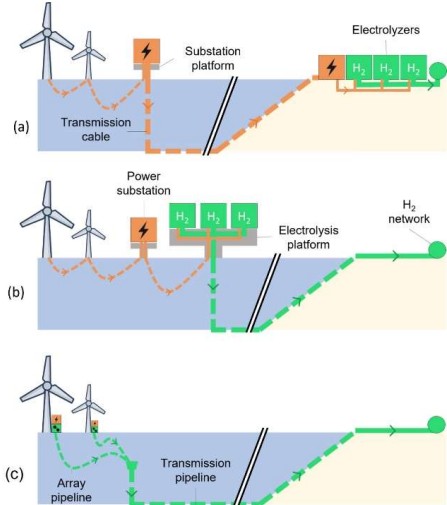

**Figure 3: Configurations considered for deployment of green hydrogen: (a) Onshore (b) Centralized offshore (c) Decentralized offshore. Reproduced from Rogeau et al. (2023) with permission from Elsevier B.V.**

In either distributed or centralised hydrogen configurations, a liquefaction unit can be integrated into the facility so that

liquified hydrogen can be produced and stored offshore to be later exported. According to Jang et al. (2022), for offshore wind

turbines, the distributed hydrogen production was found to be the most economical alternative with an LCoH2 of 13.34 $/kg$_{H2}$,

followed by the centralised (13.66 $/kg$_{H2}$) and the onshore (14.1 $/kg$_{H2}$) concepts. Ibrahim et al. (2022) extensively discusses

the advantages and disadvantages of the above configurations and also for offshore wind turbines focusing on the electrolyser

technology, the floating platform, and the energy transmission vector (electrical power or offshore hydrogen pipelines). It

suggests 'Offshore Centralised' may compete well compared with the 'Offshore Decentralised' configuration, because of

reduced complexity and greater flexibility for upgrading hydrogen equipment. However, it would have the additional capital

expenditure (CAPEX) of a floating platform to accommodate the relevant equipment and would represent a single point of

failure. In terms of total lifetime cost, Rogeau et al. (2023) report that onshore electrolysis is the most economical option,

followed by the centralised offshore and decentralised offshore schemes, but in terms of LCoH2, the offshore electrolysis

options result much cheaper, particularly for far-from-shore locations. Indeed, especially for deep waters (over a 100 m), the

decentralised option show increasingly preference over the centralised scheme for the coming years.



Bonacina et al. (2022) analysed a centralised hydrogen system assuming that a redundant oil & gas platform is reused to accommodate hydrogen production, liquefaction, and storage systems. It is concluded that a dedicated offshore-wind farm-powered system for ship refuelling purposes would be able to achieve an LCoH2 of less than 4 €/ $kg_{H2}$, depending on the electrolyser to wind capacity ratio and the wind farm power capacity. Concerning the latter, Mathur et al. (2008) have suggested that 100 MW is the minimum capacity for an economically feasible hydrogen production using offshore wind

turbines. Clearly, under the current costs of offshore wind and natural gas, these conclusions deserve to be revisited. Also, the above configurations should be investigated further by considering other ocean energy devices, as well as the integration with liquefaction and storage facilities.

## 3 Key design requirements

As for oil & gas and conventional offshore wind structures, a set of key requirements will drive the design and selection of an

offshore green hydrogen system. Some are more general and are strongly related to the techno-economics of the system, such as geographical location, resource characteristics, and the payload of the offshore system (energy production facilities, including the hydrogen system), while other design aspects are more specific to the support structure of the floating system, such as the environmental conditions, hydromechanics, structural performance of the system, and risk tolerance. The more general considerations are addressed in the following sub-sections, and those more specific to floating system characteristics

are dealt with in Section 4.

### 3.1 Site conditions and resource compatibility

For typical offshore oil & gas and renewable energy projects, the available resources, e.g., oil, gas, wind energy, wave energy etc., dictate the optimal geographical location of the offshore facility. Once the location is determined, a set of site-specific characteristics (area, water depth, distance to shore, etc.) drive the design and selection of the offshore system as a whole.

These will be certainly applicable for an offshore hydrogen system, but the availability of existing infrastructure near or at the proposed location (e.g., oil & gas platforms, depleted oil & gas reservoirs, power cables, pipelines, subsea layout, etc.) may also play an essential role in the economic feasibility of an offshore hydrogen system. Several studies have demonstrated that LCoH2 can be significantly reduced and become commercially competitive if existing facilities can be repurposed/shared. For example, Peters et al. (2020) addresses the benefits of a cost-effective, balanced and secure transition of the North Sea from

oil & gas towards renewable energy production through the PosHYdon project; Mckenna et al. (2021) considers the integration of the oil & gas and renewable energy sector for North Sea neighbouring countries and proposes the repurposing of oil & gas platforms into hydrogen production facilities that could deliver hydrogen to shore via existing and/or new purpose-built pipelines. Zulkifli et al. (2022) similarly considers the reuse of ageing fixed offshore platforms in Malaysia as OTEC power plants. Depleted oil & gas reservoirs and pipelines could also be used to store hydrogen, as suggested in Pearson et al. (2019).





Different from oil & gas, where the available resource is largely independent of the environmental conditions, green offshore hydrogen systems are inherently related to the environmental resource and therefore temporally and spatially variable. Consequently, characterisation and compatibility with energy demand is critical for the economic operation of these systems.

### 3.1.1 Metocean and geotechnical conditions

In the present context, metocean conditions refer to winds, wave, or current regimes used to characterise the environment apart
from the economic energy resource aspects, while geotechnical conditions refer to seabed characteristics and soil properties. Those conditions define the requirements for the design of the offshore system (support platform, mooring system, cables and riser systems, etc.) and dictate performance under operational and survival conditions.

### 3.1.2 Renewable energy source compatibility

Wind energy as the primary resource for offshore green hydrogen production systems should not be taken for granted based
solely on the maturity of offshore wind turbine technology. In addition to the resource potential (the theoretical amount of energy that can be exploited during a given time frame in each location), resource variability and resource predictability are key considerations for hydrogen production systems. Resource variability refers to how the resource changes in the short (hours or days) and the long (months or years) term, whereas resource predictability is the accuracy of forecasts and prediction models (D'amore-Domenech and Leo, 2019). Although resource characteristics are intrinsic to a given location, each energy
resource (wave, wind, current, solar, etc.) will exhibit independent short and long-term characteristics (for example, from season to season, year to year).

The characteristics of the available resource potential and variability should define the type(s) of technology to harness the energy from wind, wave, tidal, solar, etc., for a given site. The size, power capacity, and operational requirements of the hydrogen equipment should be then selected according to the hourly, daily, monthly and or yearly available resources.
Caglayan et al. (2019) explains that the choice of the weather year for the design of a hydrogen production system can significantly affect the total annual cost of the system. Another study (Mehta et al., 2022) concluded that designing turbines specifically for hydrogen production (LCoH2 optimised) tend to have a higher specific power over the Levelised Cost of Energy (LCoE)-optimised turbine designs, but oversizing the electrolyser compared to the turbine rating would be a better design strategy. The variability of the energy resource and its compatibility with the performance characteristics of both the
energy extraction devices and the hydrogen production equipment is a critical aspect that deserves further research.

### 3.2 Support structure "payload"

The "payload" sometimes indicated as "topside" in offshore oil & gas engineering refers to all the equipment that is not the support structure. For a hydrogen system, this equipment refers to the energy extraction devices and the hydrogen facilities.





### 3.2.1 Energy extraction devices

The devices for energy harvesting are focused on a single energy resource and usually categorised according to their working principle. Abundant literature can be found on available devices types as well as on their current development stage, e.g. Blanco-Fernández and Pérez-Arribas (2017); Irena (2020); Ittc (2017, 2021). Several of the key drivers for the selection of the energy harnessing system for offshore hydrogen are the same as for conventional electricity generation, with the most relevant being the Technological Readiness Level (TRL) and LCoE. Indeed, several feasibility studies have concluded that

LCoH2 is strongly dependent on LCoE (D'amore-Domenech and Leo, 2019; Franco et al., 2021; Raut and Goudarzi, 2018). Other characteristics such as the area and volume "footprint" (in kW/m² and kW/m³, respectively) of the device and its capacity factor also remain as essential features for hydrogen production.

On the other hand, there are additional aspects of the device that become crucial when considering the hydrogen production scenario: the available deck/internal compartments surface area and volumes, to accommodate the relevant equipment and

possibly for product storage. In the same context, another desirable feature of an energy device is scalability, i.e., the flexibility of the device to be re-dimensioned or adapted, to allow for the fulfilment of the requirements for the accommodation of the hydrogen facilities.

### 3.2.2 Hydrogen facilities

In the context of the present work, the hydrogen facilities refer to electrolysers, desalination plant, compression and/or

liquefaction machines, storage tanks, offloading machinery, pipelines, etc. i.e., all the facilities that are not present in a solely conventional offshore (electricity) energy generation system.

#### 3.2.2.1 Electrolyser technology

The choice of electrolyser technology is a primary consideration and several papers discuss and critically analyse available technologies (Buttler and Spliethoff, 2018; D'amore-Domenech and Leo, 2019; D'amore-Domenech et al., 2020; Meier, 2014).

Alkaline Electrolysis (AEL) and Proton Exchange Membrane Electrolysis (PEM) have both been reported as preferred solutions in most studies (Bonacina et al., 2022; Henry et al., 2022; Ibrahim et al., 2022; Lucas et al., 2022), with an increasing tendency to the latter due compact design, pressurised operation, load flexibility and fast response, despite being more expensive than the AEL (Buttler and Spliethoff, 2018; D'amore-Domenech et al., 2020).

Indeed, operational characteristics of the electrolyser such as power consumption, water consumption, minimum loading, or

hydrogen output, are essential aspects for the power management strategy of the whole offshore hydrogen system, i.e., the control algorithm used to allocate the available power to the different components of the system (desalination, compression and/or liquefaction units, storage system, etc.). If a decentralised system is considered, challenges arise from the fact that this equipment may be subject to displacements, velocities, and accelerations during operation, which usually do not occur in an onshore setting. The effects of motion on electrolyser performance are not well understood and need to be investigated and if



necessary, mitigated against. For an offshore hydrogen plant powered by offshore wind energy, in addition to considering the wind turbine performance as a function of average wind speed, the electrolyser (and its auxiliary equipment) performance depends on input power and its instantaneous variability. Electrolyser sensitivity or otherwise to minimum input power requirement, power intermittency and start-up time are some of the characteristics that need to be considered. From the perspective of environmental loads on the support structure, the assessment of metocean conditions is similar to other offshore
platforms.

### 3.2.2.2 Desalination unit

Both AEL and PEM electrolysers need processed water to operate (they cannot directly use seawater). In the offshore setting, the processed water is obtained from seawater using desalination units, which might be powered by the available ocean renewable energy. Distillation and reverse osmosis are the main technologies for desalination. The former produces higher
purity water (i.e., requires less post-treatment for demineralisation) but at a higher thermal energy consumption, while the latter is the most widely used technology being slightly more efficient but requiring chemical treatment before being used for electrolysis. For the analyses of offshore hydrogen systems, some authors (D'amore-Domenech and Leo, 2019; Meier, 2014) expressed a preference for distillation technology while others (Bonacina et al., 2022; Ibrahim et al., 2022) have preferred reverse osmosis. In any case, the volumetric seawater inflow and the power required for seawater pumps, compressors, and
auxiliary systems for the desalination unit should be balanced against the output power of the ocean energy device(s). Operational characteristics and component limitations of the desalination unit also need to be carefully addressed.

### 3.2.2.3 Storage and transportation

The delivery of hydrogen either as compressed gas or liquid, or in the form of another fuel, for instance, ammonia or methanol is another essential element within the offshore green hydrogen system. Each alternative has technical benefits and limitations
as well as the underlying challenge to achieve an economically feasible green fuel. Such choices directly affect the requirement for compression, liquefaction, and storage facilities. In Babarit et al. (2018) a comparison is made between compressed hydrogen (CH2) and liquefied hydrogen (LH2) concluding that CH2 scenarios have the best energy efficiency, current cost estimates for LH2 and CH2 were similar but LH2 is considered the most promising in the longer term due to expected cost reduction and much greater flexibility for delivery. According to Miao et al. (2021), the small-scale fluctuations in hydrogen
production (due to, for example the hourly or daily variability of the ocean energy resource) can be handled by a short-term storage system (e.g. buffer tanks), however, seasonal variations (potentially due to the mismatch between supply and demand) require large-scale energy management solutions. The economic feasibility of energy transportation via power cables and gas (compressed hydrogen) pipelines were investigated and concluded that, for long distances, pipeline transmission is cheaper than cables and pipelines have higher energy transmission capacity and lower energy losses. CH2 infrastructure is comprised
of gas buffer storage tanks, compression units, and gas pipelines. According to Jepma et al. (2018), a pressure of 100 bar is expected to be enough for long-distance hydrogen transportation and suitable for typical existing oil & gas pipelines.



Furthermore, if hydrogen is obtained from PEM electrolysers, the cost of compressors could be avoided or significantly reduced because the produced hydrogen based on PEM technology comes already pressurised (Ibrahim et al., 2022; Mehta et al., 2022). Another advantage of CH2 is that is that the pipelines for transportation have low operational costs and long

expected lifetimes (40 to 80 years) - Birol (2019). Indeed, according to Dnv (2022), between 50% and 80% of hydrogen pipelines are expected to be repurposed from natural gas pipelines with a cost of just 10-35% of new construction costs.

In general, exported CH2 is expected to be delivered for immediate utilisation or to be stored in large-scale facilities. Currently low utilisation and lack of energy storage capacity is one of the main reasons for the high cost of CH2. Regarding storage of CH2, some authors have proposed the use of existing offshore oil & gas pipelines (Pearson et al., 2019), underground subsea

storage considering several storage durations (Dinh et al., 2021), or the use of depleted offshore gas fields, saline aquifers or salt caverns (Scafidi et al., 2021).

On the other hand, offshore production of LH2 could be a viable alternative allowing its exportation to remote markets via ships or delivered as fuel for marine transportation (Bonacina et al., 2022). In this case, liquefaction, cryogenic storage, and offloading facilities are required and should be installed together or near the hydrogen support platform. In addition, the

offshore layout of the hydrogen system (including the ocean energy production farm) should allow for the access of shuttle hydrogen tankers to the storage platform(s). Under this scenario, a centralised typology may be an interesting option.

The transportation of hydrogen by means of alternative energy carriers such as ammonia, methanol or other liquid organic hydrogen carrier (LOHC) is another option (Birol, 2019; Henry et al., 2022; Ibrahim et al., 2022) that would require additional equipment to support, for example the methanation and/or hydrogenation processes and their products: methane and/or

methanol, respectively. The main advantage of the alternative energy carriers is the cost reduction in transportation and storage by using existing infrastructure (pipelines, ships, storage tanks, etc.).

## 4 Key support structure design considerations

Support structure considerations are specifically related to site characteristics and functional requirements, including structural reliability.

### 4.1 Metocean conditions

The characterisation and assessment of metocean conditions is perhaps the most critical aspect in the design of offshore structures. Unlike dedicated offshore energy extraction devices, where the characterisation of the energy resource (wind, wave, current, etc.) is typically done in terms of monthly, seasonal or yearly averages to obtain mean power and capacity factors, for hydrogen production systems' shorter-term characterisation is required to address the compatibility of the

instantaneous generated power with the operational characteristics of hydrogen production equipment (electrolysers, desalination units, etc.). Floating offshore wind structures need to be designed taking into account the complex structural



aerodynamics of the rotor coupled with the hydrodynamics of the floater excited by the wind, wave and machine dynamic characteristics.

In summary, similar to the design of conventional offshore floating wind energy systems, metocean conditions define the
capacity of the energy extraction devices to be installed onboard, which determines the payload of the support structure and consequently minimum primary dimensions. Other aspects such as stability, motion responses, storage capacity, or additional onboard equipment will also influence the main dimensions of the support structure and will be discussed further below.

### 4.2 Hydromechanics

From a hydromechanics perspective, for the distributed scenario, the additional hydrogen production and storage systems may
have an important impact on platform mass distribution and possibly on the wet geometry affecting stability, seakeeping and station keeping performance. In other words, the safety and operability of the whole system may be substantially impacted and therefore the primary dimensions of the platform may need to be increased. Freeboard, airgaps, motion, resonance periods, accelerations, and velocities at certain locations of sensitive equipment (e.g., wind turbine nacelle, electrolysers, separators, etc.), loads due to liquid motions inside storage and buffer tanks, interactions between platforms, risers (pipelines), cables and
mooring lines are some of the aspects that should be carefully assessed as should the survivability under extreme weather conditions. Despite their crucial importance these aspects do not appear to have been addressed yet in the literature.

### 4.3 Structural design and asset management

Currently, the majority of operational offshore wind farms are situated in the relatively shallow southern North Sea and are in the main monopile structures. As the wind industry moves to deeper seas with more powerful and larger turbines, jacket-type
support structures are becoming a strong alternative for intermediate water depths since these have a higher load-bearing capacity and offer structural redundancy that a monopile lacks. Floating substructures for offshore wind turbines are the only viable option for water depths beyond approximately 70 m (Wu et al., 2019; Veers et al., 2022). These floating support structure solutions can be grouped into four types: a) semi-submersibles, b) tension leg platforms (TLP), c) spar and d) barge. In recent years, numerous review studies have been reported on the past, present and future support structures used for offshore
wind (Leimeister et al., 2018a; Igwemezie et al., 2019; Wu et al., 2019). The design of offshore support structures are commonly conducted based on the standards prepared by the certification authorities and sector bodies such as DNV (Dnv, 2014), Abs (2020), IEC (Iec, 2009) and API (Api, 2019), combining the accumulated know-how gained by experience, engineering models supported through experimental and numerical tests.

The critical design limit states can vary depending on the type of structure employed along with the site conditions. For
example, according to Sunday and Brennan (2021), structural dynamics (avoiding resonance with blade-passing frequencies) and geotechnical issues (soil-structure interaction, scour, liquefaction, pile drivability), which are part of serviceability limit state (SLS), were invariably found to be the limiting design criterion for monopiles as opposed to the fatigue limit state (FLS) or ultimate limit state (ULS).





As far as hydrogen production and storage on the floating wind support structure are concerned, ULS and FLS can also have a significant impact on the structural design. In an attempt to design a lighter support structure with larger space to accommodate hydrogen production and storage units, the diameter-to-thickness ratio ($D/t$) of the constituent tubular members and joints is expected to increase. Consequently, with the increase of the slenderness ratio, $D/t$ local buckling phenomenon becomes more pronounced (Guo et al., 2013). Moreover, the high $D/t$ structures are much more sensitive to geometric imperfection (Li and Kim, 2022). Thus, the quantification of typical geometric imperfection (both magnitude and shape) for as-built and in-service offshore structures is necessary.

Progressive fatigue damage is a paramount concern for all dynamically sensitive offshore (wind) structures with welded-tubular joints operating under harsh offshore environments (Dong et al., 2011; Yeter and Garbatov, 2022). Under sufficient load cycles, every welded-tubular joint on an offshore structure can be a potential source of stress hotspots, leading to crack initiation, and subsequent propagation through-thickness. As far as fatigue damage assessment for offshore structures is concerned, both Time-Domain and Frequency-Domain (spectral-based) approaches are important tools in understanding criticality or otherwise.

### 4.3.1 Time-domain approach

The time-domain load analysis and structural response simulation have emerged as the recommended practice for floating and fixed offshore wind. This is only because time-domain simulations carried out by powerful numerical tools (Otter et al., 2022) can capture the coupling effects arising from wind turbine dynamics, wind-induced loading, wave-induced loadings, boundary conditions, and their interactions (Patryniak et al., 2022). Based on the design load case (power production) defined for FLS, wind-induced loading is the predominant factor for total fatigue damage, and the interaction between wave and wave-induced loading cannot be omitted, as shown by both Dong et al. (2011) and Fan et al. (2020).

The time domain approach is expected to remain the preferred solution for offshore wind structures for hydrogen production. Kvittem et al. (2011) reported that waves reduced the resonant effect of the wind-induced loading for a semi-submersible offshore wind turbine, which highlighted the nonlinear nature of the dynamic behaviour of floating wind and the importance of fully coupled analysis. Xu et al. (2019) agreed that wind-induced loading is indeed the dominating factor for fatigue consideration during operational conditions; but also pointed out that fatigue damage was found to be more sensitive to wave load, especially in high sea states, for the mooring line. Similarly, Marino et al. (2017) found a relatively low impact from waves onto fatigue loads during power production conditions, whereas a substantial underestimation was expected for the fatigue load due to the linear wave modelling during the parked condition.

Although Haid et al. (2013) and Stewart et al. (2013) were not inclined to give any conclusive recommendation in terms of simulation length for fatigue analysis, Kvittem and Moan (2015) calculated a 10-min simulation and 1-hour simulation and underestimated the 3-hour base simulation by approximately 10% and 4%. Both overestimations that come from unidirectional wind and waves as well as the underestimation that comes from omitting wave-wind-current interactions (Kvittem and Moan, 2015; Kvittem et al., 2011) should be considered.



In addition, considerable effort has been given to decreasing the computational effort required for fully coupled time-domain simulations to estimate total fatigue damage. Katsikogiannis et al. (2021) developed a lumping process to design load cases through damage-equivalent contour lines, showing a significant improvement in the computational effort by 93%, with a 6% deviation from the complete scatter analysis, which was estimated to be between 40-80% for the frequency-domain approach in Katsikogiannis et al. (2022).

### 4.3.2 Frequency-domain (spectral-based) approach

Prior to today's powerful software for the design, the time domain fatigue analysis has deemed to be too cumbersome as it deals with complex structural models of offshore platforms solved for a high number of possible load combinations. This statement is still valid as far as the multidisciplinary design optimisation problem of offshore wind turbines is concerned (Patryniak et al., 2022). Alternatively, the frequency-domain fatigue damage analysis can offer a quick and satisfactory solution to analyse and design for the ship and offshore structures depending on the loading bandwidth (Yeter and Garbatov, 2022).

A recent study reviewed the spectral methods for fatigue analysis from a historical point of view by Dirlik and Benasciutti (2021). Although other spectral methods were also discussed, the review gave more emphasis on the models developed by Dirlik and improved by Tovo–Benasciutti – regarded as the most used methods.

It has been argued in many studies that the effectiveness of the spectral methods varies depending on the relevant power spectral density characteristics (i.e., multi-mode, spectral width, close-mode etc.). Mrsnik et al. (2013) revealed that the Zhao-Baker method was more appropriate for the automotive industry spectra, whilst Yeter et al. (2016) showed that the Tunna spectral fatigue model developed for the railway industry could also be adopted in offshore wind. Nevertheless, the consensus has been that the Tovo–Benasciutti and, especially, the Dirlik method are the most effective methods to replace computationally expensive time domain solutions as far as the OWTs are concerned, as supported by Arany et al. (2014). The advantage of the success of the Dirlik method stems from the fact that it involves many bandwidth-related parameters making the model very flexible; hence, the statistical model gains higher predictive power.

In light of the studies mentioned above, it can be argued that the degree to which this overestimation manifests itself depends on the load bandwidth. For example, an offshore structure used for hydrogen production and storage is more dominantly exposed to wave and current-induced loading, and the influence of wind-induced loading might diminish for the fatigue-prone structural details below still water level, in particular.

### 4.3.3 Corrosion considerations in structural design

For effective asset management, it is imperative to have adequate models to predict corrosion severity in terms of pit depth and corrosion loss (Melchers, 2019). The knowledge acquired regarding corrosion degradation and corrosion-induced failure mechanism for ship structures, offshore structures and pipelines is almost transferable for the new generation of offshore green





hydrogen production structures as well as offshore hydrogen storage and transportation. In this context, Melchers (2019) recently gave a comprehensive review of the development of corrosion models for marine structures.

The nonlinear corrosion model presented by Garbatov and Guedes Soares (2019) accounts for the loss of corrosion protection or coating, which can also be quite useful for offshore wind support structures. Moreover, the model also includes some of the influential environmental factors such as humidity, chlorides, and temperature. Alternatively, a very realistic corrosion model is the model introduced by Melchers (2018), explaining the multi-phase nature of long-term corrosion development. On the other hand, it is a widely-accepted design assumption that uniform corrosion degradation is approximately 0.1 mm/year for

steel in the marine environment (Dnv Gl, 2016; Moan, 2018).

To be able to study the effect of corrosion, corrosion degradation needs to be modelled using probabilistic methods such as the random field model (Silva et al., 2013; Shojai et al., 2022) because there can be a large scatter in corrosion rates even within the same corrosion zone, especially in the splash zone and tidal zone as reported by Pakenham et al. (2021).

Momber (2011) pointed out that the corrosion degradation coupled with a cyclic load (corrosion fatigue) and constant load

(stress-corrosion cracking) was the most significant threat to the integrity of offshore wind structures, especially for the internal part of many legacy monopiles that were designed not to have any coating protection as reported by Hilbert et al. (2011).

### 4.3.4 Corrosion-induced failure mechanisms

Regarding corrosion, corrosion protection and control for the offshore wind industry, there is a lack of guidance as the development of current standards has relied very much on the experience in the offshore O&G industry, as also claimed by

both Duguid (2017) and Black et al. (2015). This highlights the importance of a good understanding of corrosion degradation, corrosion-related failure mechanisms, and control/mitigation strategies.

In addition to negatively affecting other failure mechanisms (Adedipe et al., 2017; Feng et al., 2020), corrosion can be the origin point of a structural integrity issue, which can occur in two distinct manners. These are corrosion pitting or slip band under cyclic load and stress corrosion cracking under sustained tensile stress, which was deemed a severe threat to the structural

integrity by Momber (2011) for offshore wind structures lacking necessary corrosion protection. In this regard, Nøhr-Nielsen (2018) identified the main concerns for the internal side of the substructure as stress corrosion cracking, hydrogen-induced stress cracking, and microbiologically influenced corrosion around the mud zone. Whereas for the external side, the durability of the coating, clashing with support vessels, distance to the anodes and high current demand were deemed to be the primary concerns.

### 4.4 Risk-based approaches

The structural integrity of offshore structures (oil & gas and offshore wind) has been discussed in multiple studies such as Amirafshari et al. (2021); Fajuyigbe and Brennan (2021); Yeter and Garbatov (2022). These reviews demonstrated a general framework for estimating the remaining life based on the reviewed technical papers. These studies primarily overlapped in describing the best practice in structural integrity assessment. However, Amirafshari et al. (2021) gave more emphasis on the



design and inspection planning optimisation, Fajuyigbe and Brennan (2021) put more effort into flaw acceptability assessment and sensitivity of failure mechanism related to the crack depth to thickness ratio, and Yeter and Garbatov (2022) focused more on retardation effect and probabilistic structural integrity assessment. This section attempts to compile best practices regarding the fitness-for-purpose assessment for offshore steel structures.

### 4.4.1 Reliability-based design, inspection, and maintenance

The probabilistic structural assessments concerning critical failure mechanisms are necessary to define the probability of failure and the failure consequence thoroughly so that an accurate risk-based assessment can be performed. In this regard, structural reliability methods can be used to incorporate the most relevant uncertainties as stochastic variables into mathematical models developed to perform the structural assessments, and the probability of failure can be calculated.

Wang et al. (2022) gave a very extensive review of the reliability analysis for OWT support structures by quantitative

probabilistic methods (First-Order Reliability Method (FORM), Second-Order Reliability Method (SORM), and Monte Carlo simulation with enhanced sampling methods). The review argued that reliability analysis had been an integral part of the design, code calibration, multi-hazard analysis, monitoring-based updating, and inspection planning for offshore wind support structures.

In addition to these quantitative probabilistic methods, Leimeister and Kolios (2018) reviewed the qualitative methods applied

in the offshore wind industry. They identified some of the challenges with reliability methods, such as dynamic system characteristics, nonlinearities, and site-specific environmental conditions leading to higher uncertainties, data confidentiality, and computational effort. The increasing need for reliability-based analyses to evaluate the structural performance accounting for uncertainties was also acknowledged by Shittu et al. (2021). For complex and highly nonlinear structural problems such as fracture mechanics-based structural integrity assessment, Shittu et al. (2021) suggested combinational approximation

methods (e.g., multi-layer artificial neutral networks and FORM) can offer an effective solution for complex, nonlinear and time-dependent structural problems such as probabilistic fracture mechanics-based structural integrity. An example of such an application was reported by Shittu et al. (2020). More details on the machine learning-based methods in structural reliability analysis can be found in Saraygord Afshari et al. (2022).

### 4.4.2 Risk-based damage tolerant framework

A reliability-based approach offers an efficient tool to assess the structural integrity of offshore structures in a probabilistic manner, which can be updated by available information obtained by in-service inspections so that necessary actions (maintenance and repair) may be planned. The inspection capability can be characterised by the probability of detection with a given confidence level (Brennan, 2013). This is because inspection quality and success depend greatly on inspectors' experience, the fidelity of non-destructive inspection technologies employed, environmental conditions, accessibility, etc.

The probability of failure and cost associated with the action to determine the optimal time to conduct an inspection, maintenance, or repair are important elements of damage-tolerant design philosophy. The overall life-cycle cost is targeted to



be minimised by developing an optimal inspection and maintenance plan. The risk-based inspection and maintenance planning for offshore wind structures has been addressed in several studies (Nielsen and Sørensen, 2021; Shafiee and Sørensen, 2019; Sørensen, 2012; Nielsen and Sørensen, 2011; Yeter et al., 2020; Thöns, 2018). Such a framework can be adopted in a more

holistic approach where the damage-tolerant design principles can be adopted from the design stage (Yeter et al., 2019; Thöns et al., 2013; Frangopol, 2011). This is even more so for integrated offshore wind to green hydrogen systems because any damage occurring in the support structure or hydrogen unit may cause a significant monetary and/or safety consequence.

### 4.5 Digitalisation Potential

#### 4.5.1 Digital twin applications for offshore support structures

The concept of condition-based maintenance combined with structural health monitoring systems (SHM), Artificial Intelligence (AI), and autonomous sensing has a vast potential within a damage-tolerant design and structural integrity framework. AI-based big data analytics offer damage detection and identification by means of pattern recognition for operators to decide which measures need to be taken and when to act to alleviate safety concerns and open opportunities for damage-tolerant design and operation.

Moreover, structural health monitoring data can be integrated with a physics-based or data-driven model of the physical asset, e.g., a finite element model in the context of structural integrity assessment. Through this integration, the counterpart of the physical asset in a digital world, i.e., the digital twin, can be developed. The digital twin has been an active research area in recent years, with various definitions given in the literature to clarify the underlying concept (Vanderhorn and Mahadevan, 2021; Tuegel et al., 2011).

In essence, a digital twin would contain the same structural conditions of the physical structures in real-time (i.e., structural configuration, detailed scantling, material property, macro and micro degradation) and can exhibit the same structural response and damage under a given scenario. Twinning would be a process of reducing the uncertainty between the physical structures and their digital counterpart. This is achieved by updating the digital counterpart using real-time monitoring data, which effectively removes the modelling assumptions. However, Liu et al. (2021) indicated that a digital twin is not a specific

technology but a concept that can be implemented with many different advanced technologies with different levels of sophistication, as categorised by Wagg et al. (2020).

In comparison with conventional structural health monitoring in which damage detection may be confined to a well-recognised pattern, the digital twin concept can be better suited to structural failure with an unknown mechanism, which may be critical for novel applications such as offshore hydrogen production. Additionally, the capability of providing structure-wide

understanding enables comprehensive health condition diagnostics and prognostics for both monitored and unmonitored structural details. This offers great advantages in the application of risk-based inspection planning for offshore structures. A key aspect of a successful risk-based inspection approach is an accurate and timely recording of the asset condition so that this information can be used effectively to make decisions about how to utilise the available inspection and maintenance resources.



Whilst the development of digital twins is centred around structural integrity management, there will be an impact on the
structural design at the initial stage because of an improved understanding for structural integrity management. Within the
context of the design of offshore floating structures, the primary aim of a digital twin is to reduce risk and uncertainties by
incorporating component/subsystem data and, where possible, shorten the testing and validation phase for the entire system
based on the assumed reduction in uncertainty (Wagg et al., 2020). Moreover, Salvino and Collette (2009) suggested that the
monitoring of real-time response and digital twin-based health condition can become an assurance of innovative design. It
would be reasonable to consider a relaxation of partial safety factors removes unnecessary structural redundancy, leading to a
more cost-effective structural design. It is also worth noting that the standardisation of monitoring is also highly relevant, as
indicated by Chen et al. (2021).

Different sensors would have different specifications such as measurement range, sampling rate, uncertainty tolerance and size
etc. In an uncontrolled environment, such as offshore, the data is subjected to a great deal of noise. A dedicated data quality
assurance is, therefore, necessary to de-risk the offshore application. In addition, prior to training supervised and unsupervised
machine learning models, big data must be cleansed from noise and outliers using digital signal processing techniques.
Currently, no guidance is available for the specification of monitoring units applied to digital twin-based structural integrity
management.

### 4.5.2 Artificial intelligence-aided structural asset management

From an offshore wind standpoint, there is a vast literature on big data collection and analysis for SHM systems, damage
identification, signal processing and machine learning. Martinez-Luengo et al. (2016) claimed that supervised and
unsupervised machine learning techniques are not only effective for pattern recognition and outlier detection (diagnostics) but
also for building predictive (prognostic) models once noisy data is treated appropriately.

In addition to machine learning algorithms, Antoniadou et al. (2015) showed that advanced signal processing techniques can
be used for vibration-based analysis for feature extraction, feature selection, dimensionality reduction and pattern recognition.
Some recommended methods to clean the noisy data are Gaussian filtering, median filtering, principal component analysis,
short-long Fast Fourier Transformation, spectrogram and least-square average, whilst nonparametric statistical analysis can be
employed to confirm the appropriateness of the SHM data preprocessing (Yeter et al., 2021).

The effort combining various monitoring techniques, advanced signal processing methods and machine learning algorithms is
also acknowledged in Stetco et al. (2019). Jiménez et al. (2019); Jiménez et al. (2018) reported a good example of combining
machine learning techniques with the wavelet transformation to detect and diagnose the delamination of the blade structure.
Provided that the sensors are optimally placed (Liu et al., 2018), structural performance can be explained based on the
acceleration responses (Gomez et al., 2013) as a result of the successful training of operational data.

The development of such complex systems is in its infancy, and there is so much room for improvement. Furthermore, the
long-term economic implications of using AI-aided structural health monitoring systems have yet to be studied thoroughly.




## 5 Discussion

### 5.1 Main knowledge gaps

This review considers that offshore wind is currently the most promising technology for offshore green hydrogen production systems. However, unlike the current shallow water monopile installations, hydrogen systems are more likely to be deployed
in relatively remote deep-water locations and for these barge and semisubmersible floaters seem to be more interesting alternatives due to their inherent available spaces that could accommodate the hydrogen process systems equipment. Current design practice and guidance for offshore wind (electrical) energy systems are not necessarily applicable to offshore hydrogen systems. For example, the characteristic that the larger the turbine rating, the lower the LCoE for offshore wind may not hold for LCoH2. Lower but less variable power inputs seem more desirable for electrolyser systems. Indeed, recent studies indicate
that dedicated hydrogen production systems are attractive over systems that only produce hydrogen from surplus electricity. Berg et al. (2021) reported that exploiting excess energy from wind farms to produce hydrogen is not cost-competitive. Moreover, according to Bonacina et al. (2022); Dinh et al. (2021), dedicated offshore hydrogen systems could fully exploit offshore wind potential because they can be installed at any appropriate location and without the need to connect to a power grid. Mehta et al. (2022) suggested that differentiated LCoH2-optimised turbines design may provide an advantage over
LCoE-optimised turbines in the future.

In terms of overall configuration, despite the available literature on typologies, necessary equipment, and costs, no studies addressing the best type of floating support platform or required adaptations on the internal arrangement of the hull and decks have been found. Hydrogen facilities also represent an increase in the payload, and likely, in the total weight of the platform thus, modifying its centre of gravity and inertia, consequently, stability, seakeeping and station keeping characteristics.
Furthermore, operational performance criteria imposed by the hydrogen production and storage equipment could be more restrictive than those of the wind turbine alone. Regarding the construction and assembly, to minimise operations at sea (e.g., heavy lift), it would be desirable to install wind turbines and hydrogen equipment before the deployment at sea. For a centralised typology, minor modifications would take place on the each of the support platforms of the wind turbines, but a new (hub) support structure would be required for the hydrogen facilities and the electrical substation where the electricity
from all the platforms is to be gathered. No studies concerning this floating hub and its particularities have been found.

The operations and maintenance strategy will also affect the design of the support structure, i.e., if the facilities are to be manned, stricter requirements will be applied regarding stability, safety, and human exposure to motions, with respect to a typical offshore wind turbine that is usually considered an unmanned system. However, even if considered unmanned, human exposure limits should be satisfied during maintenance activities (Scheu et al., 2018), and need to be further investigated along
with access systems.

Although most recent studies refer only to hydrogen either as compressed or liquified form, the choice of the hydrogen energy carrier should not be overlooked. Ammonia, methane, methanol, or other synthetic energy carriers can be produced from offshore hydrogen. Ammonia, methane and methanol, for example, have advantages on the maturity of the production



technology; the availability and affordability of transportation and storage facilities; and potential markets for industrial applications (nitrogen fertilisers, manufacturing of chemicals, plastics and textiles, mining industry, pharmaceutical, refrigeration, waste treatment, air treatment, etc. Ammonia is also being proposed as a carbon-free fuel for the maritime transportation industry (Hasan et al., 2021; Rouwenhorst and Castellanos, 2022). Therefore, if alternative energy carriers are to be considered, the necessary facilities and their operational limitations should be accounted for in the design of offshore hydrogen systems.

Structural and materials-related challenges due to the loads imposed by the hydrogen production and storage facilities on the support platform should also be carefully assessed, especially in compartments subjected to high-pressure or cryogenic temperature, sloshing impacts due to liquid motions inside tanks and compartments, corrosion, and others.

The offshore wind industry has adopted some of the standards, recommended practices, techniques and numerical tools for load and resistance analysis developed for the O&G industry, which has been beneficial but potentially limiting. This previous
know-how has allowed for substantial growth and significant cost reduction but has also resulted in knowledge gaps and misconceptions regarding offshore wind structures' design and asset management.

Given that offshore wind farms considered for hydrogen production are likely to be located in sites further offshore, with higher wind energy potential, harsher conditions are likely to be encountered, leading to higher loads. Sea states with higher wave heights can also affect the ultimate load limit state, for which the definition of design load cases becomes essential. In
this regard, for the uncertainty analysis of fatigue and ultimate loads, the partial safety factors used need to be revised because the risk associated with the offshore structure to produce or store hydrogen may be more significant than for an "unmanned" offshore wind structure.

Both fatigue and ultimate strength analyses rely heavily on the accurate definition of sea states and their statistical nature. The offshore and shipping industry has come a long way regarding environmental load statistics and predictions using state-of-the-
art remote sensing instruments such as satellites, LIDAR, etc., as well as deep learning methods. However, the industry is still in its infancy in terms of understanding the effects of climate change on short and long-term wave and wind statistics. Moreover, more caution is needed when dealing with structural response under abnormal environmental conditions, storms, rouge waves, and extreme wind gusts, which might have not only short-term effects such as plastic deformation and early crack initiation but also long-term effects on the remaining fatigue life of an offshore structure.

Since the risk for offshore hydrogen systems is more significant than for conventional offshore wind structures, uncertainty analyses of fatigue and ultimate loads are much needed to update design safety factors. The power production load cases constitute a challenging engineering problem since when the energy production is at its peak, the risk of structural stress getting near the permissible design level increases. When the structure is subjected to its limit load, it is exposed to either tension failure or buckling or both – an elastic structural analysis does not fully explain the structural behaviour under extreme loads.
Such specific events should be part of design load cases and dealt with one-way coupled (sequential) simulations.

There is substantial room for improvement in multi-physics code development integrating fully coupled aero-hydro-servo-elastic simulations with detailed structural response analysis for design optimisation. Such code developments require very





reliable wind turbine data and floaters design, whose development is more diverse, and changes happen more rapidly. The remedy for such a challenge is the collaboration involving partners from academia, research institutes, classification societies and industry.


Last but not least, digital twins are expected to pave the way for innovative structural designs and structural integrity management incorporating the latest development in structural engineering (i.e., advanced structural configuration), material science (i.e., novel material), manufacturing capability (i.e., 3D printing, additive manufacturing), or fulfilling the requirement of new operations. Likewise, mitigation to substandard design can be provided by the application of digital twins and with its

diagnostic and prognostic capabilities to increase the transparency of the consumed fatigue life and cumulative damage of critical structural locations. Preventive actions can then be taken to avoid catastrophic failure.

### 5.2 SWOT analysis: challenges and opportunities

Based on the above discussion, a SWOT (Strengths, Weaknesses, Opportunities and Threats) analysis for offshore green hydrogen systems has been performed and summarised in Table 1. Most of the challenges experienced by other offshore

industries are likely to be experienced by offshore hydrogen systems. Moreover, incorporating novel technologies or using reliable technologies in new environments should bring even more challenges. Nevertheless, it is only reasonable to argue that ever-increasing computational power (cloud-based), digitalisation (SHM, digital twin), and strong AI (supervised and unsupervised machine learning) present themselves as opportunities to deal with these emerging challenges, which were not available in the past.




**Table 1: SWOT analysis for the Design of Offshore Green Hydrogen systems**

| Strengths | Weaknesses |
|---|---|
| o Availability of reliable and computationally efficient tools for risk-based structural and hydrodynamic analyses of conventional platforms. <br> o Existing floating support platform concepts with available space to host hydrogen facilities are known and well understood. <br> o Availability of materials for offshore structures and technologies for inspection and maintenance of conventional platforms. | o Lack of standards/criteria for integrated offshore hydrogen floating systems. <br> o Lack of models for non-standard structures (e.g., Very Large Floating Structures) and hybrid configurations. <br> o Limited understanding of hydrogen degradation effects on offshore structural materials. |
| **Opportunities** | **Threats** |
| o New and emerging offshore engineering materials (composites, GFRP concrete) and structural design concepts (Very Large Floating Structures, prestressed/adaptive structures). <br> o Digitalisation - data-driven models, AI, and digital twin methods for design, construction, monitoring and inspection. <br> o Emerging developments in wind, wave, tidal and electrolyser technologies. | o Current designs of support structures for offshore wind turbines unsuitable for hydrogen facilities and/or for operational & safety requirements. <br> o Harsher environmental conditions and stricter operational constraints (associated to hydrogen production) for the support structure, mooring system and pipelines. <br> o Challenging cost requirements. |





**6 Conclusions**

A comprehensive review of offshore green hydrogen systems has been conducted from an offshore structures perspective and the applicability of recent developments in methods and tools available to support the design of these new structures. The present literature review provides a critical assessment of the knowledge gaps, challenges and opportunities, complemented by a SWOT analysis for offshore support structures for green hydrogen systems. The main conclusions are as follows:

- Distinct from offshore wind electricity generation, hydrogen systems require additional areas and volumes within support
platforms including onboard complex power management of energy production and consumption, fulfil different stability, seakeeping and station keeping limitations, and comply with potentially stricter requirements imposed by the additional equipment.

- The knowledge and experience from the offshore oil & gas and offshore wind industries can be adapted and modified for the development of reliable and resilient offshore "green" hydrogen structures.

- As an alternative to prescriptive offshore design standards, case-specific standards can offer flexibility to deal with structural design technical challenges and asset management of offshore structures, especially those at low TRL, such as emerging green hydrogen production, storage, and transportation technologies.

- Offshore wind-powered green hydrogen production is expected sited in areas with high wind energy potential, which inevitably leads to a higher environmental load for both FLS and ULS.

- Research needs to focus on the definition of relevant design load cases, projected wind and wave statistics, both short and long-term, taking into account climate change and the dynamic behaviour of floating structures under harsher sea states and corresponding nonlinear structural response.

- The time-domain load analysis and fatigue damage assessment are inherently heavily reliant upon high-fidelity simulations for the floating support structure, which increases the time required for design optimisation. Nevertheless, the spectral
method, with the aid of machine learning, might provide a useful alternative for design optimisation.

- New materials, AI, and Digital twins with diagnostic and prognostic capabilities offer a significant opportunity to be more flexible in terms of design standards and regulations, which can clear the way for innovative structural designs and O&M strategies, leading to a safe, sustainable, and value-adding life cycle of offshore support structures.

- Given the complex scenario for design, construction, installation, operation, and maintenance of the hydrogen systems as
a whole, i.e., not only the support structure and the hydrogen facilities but also the subsea equipment, the consideration of unmanned operation should be carefully considered as this can be critical for the achievement of a cost competitive LCoH2 and a safe, responsible sector.



**Competing interests**

At least one of the (co-)authors is a member of the editorial board of Wind Energy Science.

**Acknowledgements**

This work was funded by the UK Engineering and Physical Sciences Research Council (EPSRC) as part of the Ocean-REFuel (Ocean Renewable Energy Fuels) Programme Grant EP/W005212/1 awarded to the University of Strathclyde, Newcastle University, University of Nottingham, Cardiff University and Imperial College London.

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

Series, 012019,