# Peer review of "A critical review of challenges and opportunities for design and operation of offshore structures supporting renewable hydrogen production, storage, and transport"

_Wind Energy Science, 2023_

## Author Response (AR1)

**REFEREE'S COMMENTS & REPLIES**

**REFEREE 1**

**RC1**: 'Comment on wes-2023-143', Anonymous Referee #1, 20 Nov 2023
*Overall, this work encompasses an extensive and well-organized literature search. The paper can serve as a guide for a preliminary design of offshore structures for hydrogen production, despite its limited scientific contribution. In my opinion, in fact, it lacks in technical details and does not allow for quantitative comparisons. Throughout the paper, only qualitative comments are provided regarding the literature analysis conducted. Examples are reported below:*

- *pp10, line 250-253 : Alkaline Electrolysis (AEL) and Proton Exchange Membrane Electrolysis (PEM) have both been reported as preferred solutions in most studies (Bonacina et al., 2022; Henry et al., 2022; Ibrahim et al., 2022; Lucas et al., 2022), with an increasing tendency to the latter due compact design, pressurised operation, load flexibility and fast response, despite being more expensive than the AEL (Buttler and Spliethoff, 2018; D'amore-Domenech et al., 2020).*

- *pp11, line 272-274: For the analyses of offshore hydrogen systems, some authors (D'amore-Domenech and Leo, 2019; Meier, 2014) expressed a preference for distillation technology while others (Bonacina et al., 2022; Ibrahim et al., 2022) have preferred reverse osmosis.*

- *pp11, line 287-289: The economic feasibility of energy transportation via power cables and gas (compressed hydrogen) pipelines were investigated and concluded that, for long distances, pipeline transmission is cheaper than cables and pipelines have higher energy transmission capacity and lower energy losses.*

- *pp11, line 290-291: a pressure of 100 bar is expected to be enough for long-distance hydrogen transportation and suitable for typical existing oil & gas pipelines.*

**Reply:** Thanks for your valuable review and comments. As highlighted by the reviewer, the paper does not seem to provide systematic technical details and quantitative comparisons. This limitation is mainly due to the early stage of development in the renewable hydrogen sector, resulting in a scarcity of publicly available data for a comprehensive comparative study. More importantly, the primary focus of this work is to conduct a holistic system-level review, aiming to identify research gaps and the necessary developments to address them. While the inclusion of quantitative analysis and comparative studies would enhance the paper's value, there is an argument that if done for each of the aspects covered in the paper it will substantially increase its length and dilute the main message, i.e., they may somewhat obscure the clarity of the primary aim and objectives of this work, i.e. to conduct a holistic system-level review. Therefore, such in-depth analyses are more suited for subsequent works that specifically address the technical challenges of specific areas. Nevertheless, we highly appreciate the insightful comment from the reviewer and will, where appropriate,

integrate quantitative data to substantiate the qualitative statements and conclusions presented in this paper.

*Moreover, the whole section on hydrogen facilities is lacking. Storage and transport, both in centralized and decentralized offshore configurations, electrolyzer technology and desalination systems are just cited, while their impact is crucial in the design of a hydrogen plant. The issues related to storage systems on offshore platforms are superficially addressed, even though dimensions and technical limitations of the technologies are pivotal in the structure design process. An example is the mere mention of cooling systems for hydrogen liquefaction, which is addressed as the most promising technology in a long-term scenario with no further details:*

- *pp.11 line 2811-284: In Babarit et al. (2018) a comparison is made between compressed hydrogen (CH2) and liquefied hydrogen (LH2) concluding that CH2 scenarios have the best energy efficiency, current cost estimates for LH2 and CH2 were similar but LH2 is considered the most promising in the longer term due to expected cost reduction and much greater flexibility for delivery.*

**Reply**: Additional comments/discussion have been included in section 3.2, with particular emphasis on the electrolysers (section 3.2.2.1), desalination unit (section 3.2.2.2), and storage and transportation (section 3.2.2.3). Some quantitative data (and the corresponding references) have also been provided to further support our statements.

Additionally, concerning floating platforms, the issue of flexible pipelines is not adequately explored, despite their critical impact for high depths installations.

Flexible pipelines have been further discussed in section 3.2.2.3, addressing some key design considerations, interaction with mooring systems, and manufacturing, installation and operational aspects of flexible risers.

**REFEREE 2**

**RC2**: 'Comment on wes-2023-143', Anonymous Referee #2, 24 Nov 2023
The paper has a good overall approach assessing the challenges of offshore green (or renewable) hydrogen production. The first two chapters, the overview and key design parameters, present some repetitions and are more focused on offshore wind technology rather than other offshore renewable technologies. The combination of hydrogen production with different offshore renewable technologies could present different challenges and opportunity, i.e. hydrogen production system can be integrated in each floating wind turbine, however it is more difficult to imagine a H2 production system integrated in a wave energy converter. Chapter 4 and 5 are clear and well structured. The paper is more a literature review than a critical review because there is the lack of quantitative data to support conclusions.

**Reply:** Thanks for your insightful review and constructive feedback. As pointed out by the reviewer, the paper does not seem to provide systematic quantitative data and novel concepts. This is mainly due to the state-of-the-art of the offshore renewable hydrogen sector – which is still in its early development stages and mostly focus on offshore wind energy. On the other hand, the main scope of this work was to review/identify in a holistic high-level approach the various challenges that should be addressed in the design of the substructure(s) of offshore renewable hydrogen systems. It is deemed that an in-depth quantitative data analysis, conducted for every aspect covered by the review paper, would lead to a very long paper where the main aim, i.e. to conduct a holistic system-level review, is somewhat lost. Despite this, we have carefully reviewed the manuscript and, where appropriate, introduced quantitative data to further support our conclusions. We have also made the necessary modifications to improve its readability.

Regarding other marine renewable energy sources, such as wave energy converters, for hydrogen production, some proposed concepts can be found in the works of Boscaino et al. (2015); Colucci et al. (2015); Turner et al. (2009); Patterson et al. (2019); Temiz and Javani (2020). Some comments and the respective references have been included in section 2.1 of the revised version of the manuscript.

1. The paper addresses the relevant scientific question of coupling offshore wind energy (or offshore renewable energy) and hydrogen production, and in particular is focused to the analysis of offshore structure for hydrogen production that are still at the first demonstration stage.

   **Reply:** Yes. Thanks.

2. The paper is of broad international interest because offshore renewables are among the more promising technologies worldwide in the future and the coupling with hydrogen production could be a solution for mitigating the variability of the renewable production and could contribute to the development of a sustainable energetic system based on non-electrical vectors.

**Reply:** Yes. Thanks.

3.  The paper aims to present a critical review, thus there are not novel concepts. The value of the paper is in the final comments even if they are mainly qualitative comments.

    **Reply:** Yes. The aim of the paper is to present a critical review to support the design of floating substructures for offshore renewable hydrogen production rather than proposing novel concepts.

4.  The method is the literature analysis and thus can be easily reproduced, however, the critical review seems mainly based on the authors' knowledge and experience and the discussion and the conclusions are more qualitative than quantitative.

    **Reply:** Yes. The review is based and supported on scientific papers and the Authors' knowledge and experience from oil and gas, maritime and marine renewable energy sectors. As mentioned in our general reply, the manuscript reflects the state-of-the-art of the offshore renewable hydrogen sector – which is still in its early development stages, so quantitative data is scarce or not available.

5.  Assumptions are valid.  They are mainly based on journal articles and publications already reviewed.

    **Reply:** Yes. Thanks.

6.  The title is quite long and the term "green hydrogen" is not universally used hydrogen produced by renewables, in example the EU commission prefers the term renewable hydrogen https://www.europarl.europa.eu/RegData/etudes/BRIE/2023/747085/EPRS_BRI(2023)747085_EN.pdf. A suggestion could be **"A critical review of challenge and opportunity for designing offshore renewable hydrogen structures"**

    **Reply:** We agree that the title may be long, but we think it reflects quite well the aspects that we would like to highlight in our review. Also, the term "green hydrogen" is broadly recognised and used by several international institutions such as the International Renewable Energy Agency (IRENA), and documents such as the one cited in the referee's comment. Nevertheless, we have agreed in changing the title of the revised version of the manuscript to: **"A critical review of challenges and opportunities for design and operation of offshore structures supporting renewable hydrogen production, storage, and transport**".

7.  The abstract provides a concise and complete summary, however no quantitative results are reported because of the paper is a review.

    **Reply:** Yes. Thanks.

8. There is only one figure. If the readers are people from the sector is OK, otherwise some more pictures or rendering could be useful to clarify some concepts, i.e.  the different floating structure type.

    **Reply:** Thanks for your concern with readers from other sectors.  In the revised version of the manuscript we have included a figure (figure 4) to illustrate the different floating structure types.

**References**

Boscaino, V., Cipriani, G., Curto, D., Di Dio, V., Franzitta, V., Trapanese, M., and Viola, A.: A small scale prototype of a wave energy conversion system for hydrogen production, IECON 2015 - 41st Annual Conference of the IEEE Industrial Electronics Society, 9-12 Nov. 2015, 003591-003596, 10.1109/IECON.2015.7392658,

Colucci, A., V. Boscaino, V., G. Cipriani, G., Curto, D., Di Dio, V., V. Franzitta, V., Trapanese, V., and Viola, A.: An inertial system for the production of electricity and hydrogen from sea wave energy, OCEANS 2015 - MTS/IEEE Washington, 19-22 Oct. 2015, 1-10,  10.23919/OCEANS.2015.7404569,

Patterson, B. D., Mo, F., Borgschulte, A., Hillestad, M., Joos, F., Kristiansen, T., Sunde, S., and van Bokhoven, J. A.: Renewable CO2 recycling and synthetic fuel production in a marine environment, Proc Natl Acad Sci U S A, 116, 12212-12219, 10.1073/pnas.1902335116, 2019.

Temiz, M. and Javani, N.: Design and analysis of a combined floating photovoltaic system for electricity and hydrogen production, International Journal of Hydrogen Energy, 45, 3457-3469, 10.1016/j.ijhydene.2018.12.226, 2020.

Turner, M. W., Cleland, J. G., and Baker, J.: Salt Water Activated Power System (SWAPS) for ocean buoys and related platforms, OCEANS 2009, 26-29 Oct. 2009, 1-8,  10.23919/OCEANS.2009.5422338,

---

## Referee Report (RR1)

The manuscript has been improved after the revision and in my opinion can be published as it is. Due to the nature of the content (emerging sectors) and the approach (review), there is still a lack of validation data and many assumptions are general statements, however this manuscript is valuable to highlight the importance of the two sectors coupling, the challenges and the knowledge gaps to be filled in order to accelerate the development of renewable offhsore hydrogen production.